# Naphthalene vs. Benzene as a Transmitting Moiety: Towards the More Sensitive Trifluoromethylated Molecular Probes for the Substituent Effects

**DOI:** 10.3390/molecules27134173

**Published:** 2022-06-29

**Authors:** Adam Sokół, Henryk Koroniak, Marcin Hoffmann, Tomasz Siodła

**Affiliations:** Faculty of Chemistry, Adam Mickiewicz University in Poznań, Uniwersytetu Poznańskiego 8, 61-614 Poznan, Poland; adam.sokol@amu.edu.pl (A.S.); henryk.koroniak@amu.edu.pl (H.K.); marcin.hoffman@amu.edu.pl (M.H.)

**Keywords:** naphthalene, substituent effect, density functional theory (DFT), substituent effect stabilization energy (SESE), trifluoromethyl group

## Abstract

The application of DFT computational method (B3LYP/6-311++G(d,p)) to mono- and poly(CF_3_)substituted naphthalene derivatives helps to study changes in the electronic properties of these compounds under the influence of 11 substituents (-Br, -CF_3_, -CH_3_, -CHO, -Cl, -CN, -F, -NH_2_, -NMe_2_, -NO_2_, and -OH) to confront substituent effects in naphthalene with an analogous situation in benzene. This paper shows the dependencies of theoretically calculated SESE (Substituent Effect Stabilization Energy) values on empirically determined, well-defined Hammett-type constants (*σ_p_*, *σ_m_*, *R*, and *F*). Described poly(CF_3_)substituted derivatives of naphthalene are, so far, the most sensitive molecular probes for the substituent effects in the aromatic system. The presence of the trifluoromethyl groups of such an expressive nature significantly increases the sensitivity of the SESE to changes caused by another substitution. Further, the more -CF_3_ groups are attached to the naphthalene ring, the more sensitive the probe is. Certain groups of probes show additivity of sensitivity: the obtained sensitivity relates to the sum of the sensitivities of the mono(CF_3_)substituted probes.

## 1. Introduction

The substituent effect is one of the most useful concepts in modern organic chemistry. It describes how rearrangements of groups in a molecule affects its properties. The substituent effect is described by the substituent constant, *σ*, given by the equation proposed by Hammett [1]. As a model reaction in determining substituent constants, the dissociation of unsubstituted and *p*-/*m*-substituted benzoic acids was chosen. Then the substituent constant is given by Equation (1):(1)$$\sigma_{p/m}=\log K_{p/m}\left(X\right)-\log K$$
where *K_p/m_(X)* and *K* are the dissociation constants for *p*-/*m*-X-substituted and unsubstituted benzoic acids. The difference between these constants describes the effect of substituent X on the stability of the carboxylate anion. To simplify, it has been assumed that substituent effects consist mainly of two components: an inductive and a resonance. In the *para* position, both components are essential. It follows that the substituent constant can be split according to the equation:(2)$$\sigma_{p}=\sigma_{I}+\sigma_{R}$$
where *σ*_I_ and *σ*_R_ correspond to the inductive and resonance components, respectively. In the *meta* position, the resonance component is negligible because there are hardly any resonance interactions between both groups of atoms. However, identifying the Hammett constant *σ_m_* with its inductive component *σ*_I_ is only a useful simplification [2].

In recent decades, many other substituent constants have been introduced. Swain and Lupton defined parameters *F* (for field/inductive effect) and *R* (for resonance effect) and calculated them for about 200 substituents. These parameters were later normalized to be reconciled with the following dependence [2]:(3)$$\sigma_{p}\;\approx F+R$$

Most of these empirical approaches are appropriate for benzene or other aromatic systems. Substituent constants can successfully predict the rate constants and equilibrium constants of various groups of reactions. However, Hammett-like approaches have limits. The value of the substituent constant depends on the solvent. Moreover, some benzoic acid derivatives are insoluble in water, so measuring dissociation constants is impossible. More complex derivatives could also be hard to synthesize. Nowadays, quantum-chemical methods are used to study the dependence of compound properties on molecular structure. Many parameters, such as electrostatic potentials [3], HOMO/LUMO energies [4], energy decomposition analysis (EDA) [5], or charge of the substituent active region (qSAR) [6], correlate well with Hammett constants *σ*. One of the more important parameters is SESE (substituent effect stabilization energy), based on a homodesmic reaction approach [7,8,9,10,11,12]. SESE is calculated for the hypothetical homodesmic reaction:X-R-Y + R → X-R + R-Y(4)
as follows:SESE = *E*_*X*-*R*_ + *E*_*R*-*Y*_ − *E*_*X*-*R*-*Y*_ − *E*_*R*_(5)

SESE describes how two substituents *X* and *Y* interact with each other. The substituent *Y* could be a probe group, while *R* acts as a transmitting moiety. In molecule X-R-Y, the two substituents interact with other as they are attached to the same ring. On the right side of the equation, they are connected with different rings, so they are not interacting. If both substituents are electron-donating groups (EDG) or electron-withdrawing groups (EWG), they both push or pull electrons, so it will be preferable to have these substituents attached to two different rings. Consequently, the enthalpy of the homodesmic reaction will be negative. On the other hand, if one of the substituents is EWG and the second is EDG, they do not ‘compete’ and the enthalpy will be positive. It has been proven that the traditional substituent constants could be replaced by the SESE to estimate the substituent effect [9,10,11,12].

Research on the substituent effect in benzene has a long story, but there are hardly any discoveries in the case of naphthalene [13]. Naphthalene is an interesting transmitting moiety in exploring substituent effects because there are five times more possible disubstituted naphthalene derivatives (*meta*- or *para*-type) than in the case of benzene (Figure 1 and Figure 2).

In contrast to benzene, naphthalene could be substituted with more than one conjugated probe group (putting aside the *ortho*-substitution). It gives us a chance to examine a higher number of structures and interactions between substituents. Possible structures of disubstituted naphthalenes are shown in Figure 1 and Figure 2. In *para*-type structures, interactions between donor and acceptor are inductive and resonance while in *meta*-type structures interactions are mostly inductive. Derivatives marked as red have different mutual orientation of substituents. It can be important in terms of e.g., field effect.

Molecules with fluorine atoms are useful as probes of the substituent effect. This chemical element affects the chemical properties of compounds due to its high electronegativity (almost 4.0 on the Pauling scale), which causes a strong withdrawing inductive effect. Hence, the fluorine atom acts as a *σ*-electron acceptor. On the other hand, the fluorine atom is a π-electron donor due to the resonance effect caused by its lone-pair electrons. While fluorine has a ‘double nature’, perfluoroalkyl groups are always electron-withdrawing.

Due to its high electronegativity, the presence of a fluorine atom causes a tremendous change in the electron density of the molecule. On the other hand, a fluorine atom is only slightly larger than a hydrogen atom, so replacing hydrogen with fluorine does not lead to large changes in the shape of a molecule [14]. The presence of fluorine atom or fluoroalkyl group causes changes in acidity, the strength of hydrogen bonds, lipophilicity, and other physicochemical and biological properties of the compounds [15]. Fluorine plays a special role in medicinal chemistry, where the replacement of one atom with another leads to significant differences in biological activity. Fluorinated compounds can be less or more selective, more efficient, or easier to administer. There are plenty of drugs containing fluorine atom or trifluoromethyl group (CF_3_). These include anxiolytics (e.g., midazolam [16]), antidepressants (citalopram [17], fluvoxamine [18], fluoxetine [19]) and antipsychotics (e.g., haloperidol [20], flupentixol [21], risperidone [22], and fluphenazine [23]).

Let’s compare four substituents: CH_3_/CH_2_F/CHF_2_/CF_3_. The two middle ones are the partially fluorinated methyl groups emerging in medicinal chemistry in tuning electronic properties. All of their Hammett constants are *σ_p_* = −0.17/0.11/0.32/0.54, respectively [2]. Increasing values mean more electron-withdrawing character due to the larger number of fluorine atoms. This shows how the effect of fluorine is “evenly” distributed from CH_3_ to CF_3_. The same can be seen in the SESE values coming from the interactions of these four substituents with e.g., the NO_2_ group in *p*-substituted benzene derivative **X-(*p*-Ph)-NO_2_** where **X** = CH_3_/CH_2_F/CHF_2_/or CF_3_. The calculated SESE values are 0.8/−0.3/−1.3/−2.2, respectively. Here, lower values indicate more electron-withdrawing character of X. Moreover, the determination coefficient in the correlation between these values (SESE vs. *σ_p_*) is very high, *R*^2^ > 0.99. It is a simple demonstration that the substituent effect estimated by SESE displays comparable effectiveness as the traditional substituent constants, *σ_p_*.

It has been reported [24,25] that SESE values for *m*-/*p*-substituted benzene derivatives with a fluorine atom or trifluoromethyl group correlate well with corresponding Hammett constants. Moreover, the CF_3_ group is a sensitive probe for exploring the substituent effect in the case of benzene. The more CF_3_ groups are attached to the ring, the more sensitive probe is [25]. The sensitivity is a very important parameter due to the accuracies of the computed energies (0.5–1.0 kcal mol^−1^) [26]. Thus, increasing the sensitivity of the probe decreases uncertainty in the estimation of the substituent constants. Two or three CF_3_ groups cause higher sensitivity due to the nature of this substituent. Several electron-withdrawing groups strongly affect the other EDG or EWG group attached to the benzene ring. Therefore, changes in the substitution significantly affect the SESE [25].

The motivation of this report is to show that: (a) SESE and *σ* values correlate also in the case of naphthalene and (b) if polysubstituted trifluoromethyl derivatives of naphthalene are more sensitive probes for the substituent effect.

## 2. Methodology

Molecular geometries of 29 types of naphthalene derivatives were optimized using Density Functional Theory at the B3LYP/6-311++G(d,p) level of approximation (see Figure 1) [27,28,29]. This method was chosen as the one which was described to give fine results in studies on the substitution effect [11,25,30]. The frequency analysis was performed at the same level of theory to verify if the optimized geometries correspond to the stationary points. All the calculations were done using Gaussian 09 [31].

In this paper, the examined derivatives are described by symbols: P_1_-P_2_ (P_1_ is a position of a substituent (X), P_2_ are positions of trifluoromethyl groups; for simplicity, the substituent X has always attached number 1 or 2). In all the groups there were 12 compounds with one of the substituents shown in Table 1 [2,24].

## 3. Results and Discussion

Table 2 presents values of the slope, *a*, and determination coefficient, *R*^2^, for the dependence of SESE values on *σ_p_*, *σ_m_*, SESE **X-(*p*-Ph)-CF_3_**, SESE **X-(*m*-Ph)-CF_3_**, *R* and *F* values which were taken from the previous work [2].

### 3.1. Dependence of SESE Values of Analysed Probes on SESE Values of X-(p-Ph)-CF_3_

From all the mono(CF_3_)substituted derivatives, structure **2**-**6** is the one with the highest value of *R*^2^ = 0.99. The substituent effect’s transmission in structure **2**-**6** is, therefore, very similar to that in *p*-trifluoromethyl-X-benzene (see Figure 2).

Most of the *para*-type mono(CF_3_)substituted structures show better correlations than *meta*-type ones due to a similar type of substitution like in the case of **X-(*p*-Ph)-CF_3_**. On the other hand, the worst correlation shows **2**-**7** (*R*^2^ = 0.80), which can be treated as *m*-disubstituted benzene with an additional ring (see Figure 3). Due to different type of substitution, values of SESE of these structures does not correlate well with SESE of **X-(*p*-Ph)-CF_3_**. Similarly, in the group of di(CF_3_)substituted derivatives, the worst correlation shows **1-36** (*R*^2^ = 0.77), the probe with two trifluoromethyl groups in *meta* positions.

Let us compare two structures: **1**-**5** is similar to **2**-**6** because in both structures substituent X and the CF_3_ group are divided by four carbon atoms in the molecule. However, **1**-**5** has one of the worst determination coefficients of all the examined structures (*R*^2^ = 0.86). Structure **2**-**6**, in some way, resembles benzene—both substituents are in β positions, so they are surrounded by secondary carbon atoms. The structure **1**-**5** is different–substituents are in α positions and are close to tertiary carbon atoms. It can be seen also in the case of **1**-**6** (*R*^2^ = 0.92) and **2**-**5** (*R*^2^ = 0.98): β-X-substituted derivative has a higher value of *R*^2^. It can be concluded that structures with substituents in β positions are better probes of the substituent effect.

Structures **1**-**4** and **2**-**6** should also be compared. They both resemble *p*-(trifluoromethyl)benzene. Their sensitivities differ drastically (*a* = 1.00 for **1**-**4**; *a* = 0.71 for **2**-**6**). The reason is that in **1**-**4**, two substituents are closer than in **2**-**6** and, therefore, they interact more. As a result, SESE of **1**-**4** type structures changes more noticeably than in the case of **2**-**6**. According to that, it can be concluded that in terms of sensitivity, **1**-**4** substitution better resembles *para*-substituted benzene than **2**-**6**. On the other hand, **2**-**6** (*R*^2^ = 0.99) shows a slightly better correlation than **1**-**4** (*R*^2^ = 0.95). It is probably related to the steric hindrance in position 4, close to the additional ring.

From all the examined structures, the best correlations (*R*^2^ > 0.99) show **2**-**6** and **2-5678** due to structural similarity: the first resembles *p*-X-(trifluoromethyl)benzene, whose SESE values are independent variables in this relationship; the second resembles *p,m*-substituted benzene with three substituents: the X and “tetra-CF_3_-alkadienyl” group occupying both *para* and *meta* positions.

Analyzed molecules can be divided by the types of substitution. The highest precision of correlation from all the *para*-type structures characterizes structures **2**-**6** and **2-5678**, mentioned above. In the group of *m*-substituted derivatives, structure **2**-**5** demonstrates the best correlation (*R*^2^ = 0.98). Determination coefficients of *meta*-type derivatives (*R*^2^ < 0.87 for some structures) are rather worse than for *para*-type ones due to a different type of substitution than in *p*-X-(trifluoromethyl)benzene. Among the structures substituted both in *para* and *meta* positions, it can be observed that the highest determination coefficients show tris- and tetrakis(trifluoromethyl)naphthalenes with the substituent *X* in position 2 and the CF_3_ groups in positions 4 and 5 (*R*^2^ > 0.96). Surprisingly, the correlation in the case of structure **2-45** (with only two CF_3_ groups) is slightly worse (*R*^2^ = 0.95).

The highest sensitivity, measured by the value of the slope, can be obtained in the case of penta(CF_3_)substituted derivatives: **1-34567** (*a* = 2.72) and **2-45678** (*a* = 2.78). Figure 4. shows selected diagrams comparing dependences between SESE of substituted naphthalene and *p*-substituted benzene for similar derivatives with the increasing number of the CF_3_ groups in the molecule. This comparison shows that sensitivity increases with the number of the trifluoromethyl groups. In the group of *p*-mono(CF_3_)substituted derivatives, **1**-**4** has the highest value of the slope (*a* = 1.00) while **1-3** is the most sensitive *m*-mono(CF_3_)substituted structure (*a* = 0.73). It is because the trifluoromethyl group and the substituent are closer, so the CF_3_ group strongly affects the substituent. Generally, derivatives with the substituent X and the CF_3_ group close to each other are more sensitive probes than the ones with two more distant substituents. Still, values lower than 1 (*a* < 1) indicate that those probes have lower sensitivity than the corresponding *p*- or *m*- substituted benzene derivatives.

In many examples of probes with several CF_3_ substituents, the obtained sensitivity (*a*) relates to the sum of the sensitivities of the corresponding mono(CF_3_)substituted probes. It is worth noting, however, that in the case of the dependency of SESE values on SESE values of **X-(*p*-Ph)-CF_3_**, only certain groups of probes show a kind of synergism in additivity of sensitivity. Table 3 presents obtained values of the slope and values which are sums of parameters *a* for corresponding mono(CF_3_)substituted derivatives, *a_s_*. For most of the probes the relative error, *η*, is negative (no synergism). Only for structures substituted by trifluoromethyl groups in positions 4 and 5, the value of *η* is positive. The majority of di(CF_3_)substituted probes and **1-457** show synergic additivity of the sensitivity. The sum of the sensitivities of the corresponding monosubstituted derivatives is consistent with the sensitivity of the di- or tri(CF_3_)substituted derivative. Only in the cases of **1-36** and **2-45** is this effect not observed, although the deviation from additivity is of the opposite nature. Structure **1-36** has separated substituents that interact less with each other than might be expected. Consequently, the actual value of the slope is lower than anticipated. On the contrary, the value of *η* for **2-45** is positive and the slope is more than a fifth greater than the predicted value. Moreover, **1-45** and **1-457** are characterized by positive values of *η*. In the group of tetra(CF_3_)substituted derivatives all the values of *η* are negative, but only for **X-45NM** (X = 1 or 2; N, M = 6, 7, or 8) the approximate additivity can be observed; the absolute value of the relative error is less than 10%. Structures with the CF_3_ groups in positions 4 and 5 have deformed aromatic rings due to the steric hindrance (see Figure 5). For this reason, the resonance component in the interaction between the substituent X and trifluoromethyl groups is less significant than in planar rings. This allows the conclusion that either the resonance effect has a negative impact, or the deformation itself has a positive impact on increasing the sensitivity of the substituent effect probes.

### 3.2. Dependence of SESE Values of Analysed Probes on SESE Values of X-(m-Ph)-CF_3_

The highest precision (the highest value of *R*^2^) has structure **1-3** (*R*^2^ = 0.992), which resembles *m*-trifluoromethyl-X-benzene with an additional ring (see Figure 6). Furthermore, **1**-**6** (*R*^2^ = 0.986), similar to the *m*-substituted benzene (see Figure 7), gives a satisfactory value of the correlation coefficient. However, the sensitivity is lower due to the more distant substituents (*a* = 0.99 for **1-3** and 0.61 for **1**-**6**). Surprisingly, the worst correlation shows **1-36** (*R*^2^ = 0.82), which is a *meta*-type structure with three distant substituents. The low value of *R*^2^ indicates disturbed interaction between substituents.

The highest sensitivity (the highest value of the slope) once again characterizes penta(CF_3_)substituted derivatives: **1-34567** (*a* = 3.64) and **2-45678** (*a* = 3.59). Figure 8 shows that an increasing number of the CF_3_ groups has a positive effect on sensitivity. In the group of mono(CF_3_)substituted derivatives, **1**-**4** has the highest value of the slope (*a* = 1.28), although it is not a *meta*-type structure. The higher value of the slope is, in this case, the result of a wider range of variation for the values of *σ_p_* than *σ_m_*. The most sensitive *m*-mono(CF_3_)substituted structure is **1-3** (*a* = 0.99). Still, a value lower than 1 (*a* < 1) indicates lower sensitivity than the corresponding *m*-substituted benzene probe **X-(*m*-Ph)-CF_3_**. Generally, the further away the CF_3_ group and the X group are, the lower the sensitivity of the probe is. It is worth noting that in the case of relation of SESE of **X-(*m*-Ph)-CF_3_** all the probes have higher slope values than in the case of the previous relation of SESE of **X-(*p*-Ph)-CF_3_**.

In Table 4, obtained values of the slope and values which are sums of parameters *a* for corresponding mono(CF_3_)substituted derivatives are shown. Generally, the relative error of *a_n_* is negative. It is positive (synergism in additivity) for two di(CF_3_)substituted kinds of structures: **1-45** and **2-45**. They are both substituted by trifluoromethyl groups in positions 4 and 5; the closure of the CF_3_ groups deforms the ring and disturbs the effect of additivity. The relative error is also positive for **1-457** (it has also the CF_3_ group in positions 4 and 5), but it is lower than the previous ones. Other poly(CF_3_)substituted derivatives with occupied positions 4 and 5 have negative relative errors.

As in the case of dependence on SESE[**X-(*p*-Ph)-CF_3_**], the additivity of the sensitivity of di- and tri(CF_3_)substituted derivatives could be observed. Once again, only **1-36** and **2-45** do not follow this trend. From the tetra- and penta(CF_3_)substituted probes only structures with four CF_3_ groups, substituted in positions 4 and 5, show the additivity of the sensitivity. Obtained values of relative errors of parameter *a_n_* are very similar to the ones in the previous section.

### 3.3. Dependence of SESE Values on σ_p_ and σ_m_

Structures with the highest precision (*R*^2^ > 0.97) are: **2-5678** (*R*^2^ = 0.99), **2**-**6** (*R*^2^ = 0.99) and **2-4568** (*R*^2^ = 0.97) (correlation with *σ_p_*); **2-4** (*R*^2^ = 0.98) (correlation with *σ_m_*). **2-5678** has the highest precision of all examined structures. Structures of the highest precision in correlation with *σ_p_* usually have more than two CF_3_ groups. For correlation with *σ_m_*, the effect is reversed: structures of the higher precision are the *meta*-type ones with substituent X and the one or two CF_3_ groups close to each other. A low number of substituents causes less imposing interactions between them. For *m*-mono(CF_3_)substituted derivatives structures with higher precision are the ones with the shortest distance between X and CF_3_. Comparing *para*-type and *meta*-type derivatives it can be stated that generally *para*-type structures correlate well with *σ_p_* and *meta*-type structures–with *σ_m_*, obviously.

In the group of *p*-substituted derivatives, the best correlation with *σ_p_* characterizes structures **2**-**6** (*R*^2^ = 0.99) and **2-68** (*R*^2^ = 0.96), which are 2-X-substituted with a long distance between substituent X and trifluoromethyl groups; moreover, both structures have the CF_3_ group in position 6. In the case of correlation with *σ_m_* (for *para*-type structures), **1-7** (*R*^2^ = 0.94) and **1-57** (*R*^2^ = 0.93) have the highest *R*^2^ values; both structures are 1-X-substituted, have substituents far from each other and have the CF_3_ group in position 7. Similar observations can be made in the group of *m*-substituted derivatives in case of correlation with *σ_p_*: the best are **2**-**5** (*R*^2^ = 0.96) and **2-45** (*R*^2^ = 0.90), both 2-X-substituted with the CF_3_ group in position 5, but the substituents are closer. For correlation with *σ_m_ meta*-type structures with the highest precision are **2-4** (*R*^2^ = 0.98) and **1-36** (*R*^2^ = 0.89). In these molecules, substituent X and the CF_3_ group are close, divided only by one carbon atom. In derivatives, *p*- and *m*-substituted (“mixed”, see Table 2) dependencies are more complex, but it can be observed that better correlations with *σ_p_* show derivatives with the substituent X in the β position, while better correlations with *σ_m_* describe structures with the substituent X surrounded by the CF_3_ groups (divided by one carbon atom).

α- and β-mono(CF_3_)substituted derivatives can be also compared. As in the case of “SESE-SESE” dependence, described above, **2**-**6** (*R*^2^ = 0.99) has a higher determination coefficient (correlation with *σ_p_*) than the similar structure **1**-**5** (*R*^2^ = 0.84). It can be stated again that β-substituted derivatives are better probes of the substituent effect than α-substituted ones.

The highest value of the slope (for correlations with both *σ_p_*/*σ_m_*) from all the mono(CF_3_)substituted derivatives have structures: **1**-**4** (*a* = −2.56/−4.31) and **2-8** (*a* = −1.98/−3.42) (in the group of para-type structures); **1-3** (*a* = −1.89/−3.44) and **2-4** (*a* = −1.51/−2.91) (in the group of meta-type structures). In these molecules, CF_3_ and X groups are quite close to each other. As is stated in the previous chapter, the closest the CF_3_ group and the substituent are, the more sensitive the probe is, although this is a tendency rather than a linear relationship.

The values of the slope for each structure are greater for the correlation of SESE with *σ_m_* than with *σ_p_*, even for derivatives with the CF_3_ group in the para position. Moreover, it can be observed that the more CF_3_ groups have been used, the higher sensitivity can be achieved. The highest sensitivity characterizes penta(CF_3_)substituted structures: **1-34567** (*a* = −7.07/−12.54) and **2-45678** (*a* = −7.17/−12.42). Figure 9 shows that the slope is the greatest for penta(CF_3_)substituted derivatives.

For correlation with *σ_p_*, it can be observed that in the case of tetra/penta(CF_3_)substituted derivatives, the β-X-substituted have higher precision than α-X-substituted. For correlation with *σ_m_*, tetra/penta(CF_3_)substituted derivatives have worse determination coefficients than mono- or di(CF_3_)substituted.

In the case of di- and tri(CF_3_)substituted derivatives, additivity of their sensitivities can be observed (Table 5). The absolute value of relative error, *η*, of *a*_s_ does not exceed 7%. Only in the case of **2-68** is the absolute value is more than 13% (−13.27%). For structures **1-45**, **1-457,** and **2-45**, the relative errors have positive values. Once again, it can be explained by the deformation of the aromatic rings in the molecule with two CF_3_ groups close to the ’bridgehead’ carbon atom, as explained in the previous chapter. For other structures, the effect is weekly synergistic.

For tetra- and penta(CF_3_)substituted structures, the model of additivity of sensitivities doesn’t work. For most of these derivatives relative error is lower than −10%. These structures have the CF_3_ groups both in *para*- and *meta*-type positions so their influences on substituent X partly weaken each other. For this reason sensitivities of tetra- and penta(CF_3_)substituted naphthalene derivatives are lower than the sums of sensitivities of mono(CF_3_)substituted ones.

### 3.4. Dependence of SESE Values on F and R

SESE values of analyzed probes correlate poorly with resonance (*R*^2^ < 0.8) and field (*R*^2^ < 0.86) constants. As the number of CF_3_ groups increases, the sensitivity of the probes also increases, which is reflected in the rise of values of the slope (*a* = −8.24 for **2-45678**, correlation with *R*; *a* = −17.79 for **1-34567**, correlation with *F*). Nevertheless, neither probe can be considered suitable for estimating the values of the *R* and *F* constants.

It is worth noting, however, that in the case of the dependence of SESE on the constant *R*, there is a group of substituents (CF_3_, CH_3,_ CHO, CN, H, and NO_2_) for which the SESE values deviate from the linear regression. The rest of the tested substituents are resonance π-electron-donating (Br, Cl, F, NH_2_, NMe_2_, OH). Probes containing only these substituents most often show correlations between SESE and *R* constant with satisfactory accuracies. In this case, there are several probes with a correlation coefficient *R*^2^ > 0.95; the highest precision has structure **1-57** (*R*^2^ = 0.97) with substituents spaced apart and, thus, with reduced inductive interaction between them. The most sensitive probe is **2-45678** (a = −10.64).

On the contrary, if the dependence of SESE on the constant *F* is analyzed and the resonance electron-donating groups are omitted, it turns out that all probes show a good correlation (*R*^2^ > 0.92). Although the absolute values of the slope are lower than in the case of the correlation with all tested substituents, the correlation coefficients increase significantly. Once again, the best correlation is shown by **1-57** (*R*^2^ = 0.99). Some mono(CF_3_)substituted probes also have high precision: **1-3**, **1**-**6**, **1**-**5** (*R*^2^ = 0.98) and **2-8** (*R*^2^ = 0.9794). Polysubstituted trifluoromethyl derivatives show a slightly lower precision, but their sensitivity is much higher than that of the mono(CF_3_)substituted probes. The most sensitive probe for this kind of dependence is **1-34567** (a = −13.15, *R*^2^ = 0.95). Among the tetra(CF_3_)- and penta(CF_3_)substituted probes, noticeably higher sensitivity values are shown by those with trifluoromethyl groups in positions 3, 4, and 7.

The results on the sensitivity additivity of the relationship between SESE and the *R* and *F* constants are quite interesting. Only relationships for a limited number of substituents will be discussed below, as described above (see Table 6). For the dependence of SESE values on *R*, additivity can be observed, surprisingly, for most of the probes, even for tetra(CF_3_)- and penta(CF_3_)substituted, which mostly do not show additivity of sensitivity for other types of analyzed dependences. The additivity model gives the best results for probes **2-68** (*η* = 0.35%) and **2-45678** (*η* = −0.41%)–the sum of slope values for mono(CF_3_)substituted derivatives corresponds precisely with the slope value for poly(CF_3_)substituted probe. However, there are several probes (**1-36**, **1-45**, **2-45**, **1-3457**, **1-3467**, **1-3567**) that do not show additivity. For the relationship of the SESE on *F*, most of the probes show an anti-synergistic effect of additivity. Sensitivities of di(CF_3_)- and tri(CF_3_)substituted derivatives fit in the additivity model with the exception of **2-45** (*η* = 14.05%) which exhibit synergy. The values of the slope for the tetra(CF_3_)- and penta(CF_3_)substituted probes deviate significantly from the sums of the sensitivities of corresponding monosubstituted probes (*η* = 15–30%).

## 4. Conclusions

In this paper, the newly discovered sensitive and precise probes of the substituent effects were discussed. The observed dependences confirm the statement that polysubstituted trifluoromethyl derivatives of naphthalene work well as probes for those effects. The studies allowed us to come to the following conclusions:In the case of *para*-type substituted naphtalenes, structure **2**-**6** (*R*^2^ = 0.99) is the one with the highest similarity to the *p*-substituted benzene ring measured by the value of *R*^2^. On the other hand, structure **1**-**4** is the one with the closest sensitivity to the *p*-substituted benzene ring measured by the value of the slope (a = 1.00). The substituent effect’s transmission in structures **1**-**4** and **2**-**6** is, therefore, very similar to that in *p*-trifluoromethyl-X-benzene;In the case of *meta*-type substituted naphtalenes, structure **1-3** is the one with the highest similarity to the *m*-substituted benzene ring measured by the value of *R*^2^ as well as the value of the slope (*R*^2^ = 0.99 and *a* = 0.99). From the structures having groups X and CF_3_ substituted to different rings, structure **1**-**6** has the highest similarity to the *m*-trifluoromethyl-X-benzene in terms of accuracy and sensitivity;It can be stated that β-substituted derivatives are better probes of the substituent effects than α-substituted ones and the sensitivity of such molecular probes increases with the number of the substituted trifluoromethyl groups;In the case of correlations of the SESE values with the resonance (*R*) or inductive (*F*) component of the substituent effect, it is better to correlate separately the π-electron-donating substituents and the non-π-electron-donating ones;Based on the above research results, a few new precise molecular probes with high sensitivity can be considered for future usage, especially those marked as **2-5678**, **2-4568**, and **2-45678**, among which coma the last one is the most sensitive SESE probe obtained so far. The more sensitive response of SESE values on the substitution means more accurately determining the electron-donor/acceptor nature of the examined substituent. It is a step forward toward the more sensitive molecular probes for the substituent’s effects.

## Data Availability

The data are available from the corresponding author.

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
