# Peer review of "Naphthalene vs. Benzene as a Transmitting Moiety: Towards the More Sensitive Trifluoromethylated Molecular Probes for the Substituent Effects"

_molecules, 2022, doi:10.3390/molecules27134173_

Round 1
Reviewer 1 Report
this is an article by T. Siodla and Col. where the electronic properties of different mono- and poly(CF3)substituted naphthalenederivatives have been studies by DFT computational method reaching some conclusions, somehow already expected based on the nature of the studied compounds.
From my point of view, it deserves to be published in Molecules once the following considerations have been taken into account.
A general question that should be clarified is the use of significant figures in the slopes “a”. The precision should be clarified: slopes “a” have 4 decimal units. Where the selection of four decimal units comes from? Is it meaningful to compare tables of 2 decimal units (Table 1) and others with 4 (i.e. Table 2).
One of the questions that arises and would enhance the paper is what the expectation of groups is such as CHF2 and CH2F in the mono-substituted probes. Considering the emergence of this partially fluorinated groups in medicinal chemistry in tuning electronic properties it would be nice to have at least an example covering these. Would they be evenly distributed between CH3 and CF3 SESE?
Line 319, page 7: “in derivatives p- and m-“. Revise odd typography in meta.
Line 344, page 7: “tet-ra/penta”. It should say tetra.
Line 480, Reference 15, it should be corrected with the journal in italic.
In general images resolution should be enhance
Author Response
Dear Reviewer,
Thank you very much for the positive assessment and valuable comments provided. We have addressed all the comments as shown in the revised manuscript.
Sincerely, on behalf of all authors
Tomasz Siodła
In response to your comments:
- “A general question that should be clarified is the use of significant figures in the slopes “a”. The precision should be clarified: slopes “a” have 4 decimal units. Where the selection of four decimal units comes from? Is it meaningful to compare tables of 2 decimal units (Table 1) and others with 4 (i.e. Table 2).”
All discussed values of slopes “a” and determination coefficients “R2” have been rounded to two decimal places.
- “One of the questions that arises and would enhance the paper is what the expectation of groups is such as CHF2 and CH2F in the mono-substituted probes. Considering the emergence of this partially fluorinated groups in medicinal chemistry in tuning electronic properties it would be nice to have at least an example covering these. Would they be evenly distributed between CH3 and CF3 SESE?”
The following paragraph has been added to the introduction (new line 110):
Let’s compare 4 substituents: CH3 / CH2F / CHF2 / CF3. Two middle ones are the partially fluorinated methyl groups emerging in medicinal chemistry in tuning electronic properties. All of their Hammett constants are σp = -0.17 / 0.11 / 0.32 / 0.54, respectively [2]. Increasing values mean more electron-withdrawing character due to the larger number of fluorine atoms. This shows how the effect of fluorine is “evenly” distributed from CH3 to CF3. The same can be seen in the SESE values coming from the interactions of these 4 substituents with e.g. the NO2 group in p-substituted benzene derivative X-(p-Ph)-NO2 where X = CH3 / CH2F / CHF2 / or CF3. The calculated SESE values are 0.8 / -0.3 / -1.3 / -2.2, respectively. Here, lower values indicate more electron-withdrawing character of X. Moreover, the determination coefficient in the correlation between these values (SESE vs σp) is very high, R2 > 0.99. It is a simple demonstration that the substituent effect estimated by SESE displays comparable effectiveness as the traditional substituent constants, σp.
- “Line 319, page 7: “in derivatives p- and m-“. Revise odd typography in meta.
Line 344, page 7: “tet-ra/penta”. It should say tetra.
Line 480, Reference 15, it should be corrected with the journal in italic.”
All editing errors have been tracked and corrected.
- “In general images resolution should be enhanced.”
The low resolution of figures have been enhanced.
Moreover, images and tables have been replaced alongside the text to fulfill the pages.
We have also added or edited the following sentences:
New line 162:
“From all the mono(CF3)substituted derivatives, structure 2-6 is the one with the highest value of R2 = 0.99. The substituent effect’s transmission in structure 2-6 is therefore very similar to that in p-trifluoromethyl-X-benzene (see Figure 2)”
New line 192:
“(…)it can be concluded that in terms of sensitivity, 1-4 substitution better resembles para-substituted benzene than 2-6”
New line 261:
“However, the sensitivity is lower due to the more distant substituents (a = 0.99 for 1-3 and 0.61 for 1-6).”
New line 263:
“The low value of R2 indicates disturbed interaction between substituents.”
We hope you will find our new manuscript to be sufficient for publication. Thank you!
Reviewer 2 Report
The paper describes DFT studies on mono- and poly(CF3)substituted naphthalene under the influence of 11 substituents (-Br, -CF3, -CH3, -CHO, -Cl, -CN, -F, -NH2, 10 -NMe2, -NO2, -OH), to be compared with substituted benzene. I have only a few minor comments:
- In Schemes 1 and 2, there are structures in red, for which it is not obvious to consider them as para- (Scheme 1) or meta-derivatives (Scheme 2). An explanation is the text is needed.
- In the captions to Figures 4, 8, and 9, it should be indicated that the comparison if made with benzene derivatives
- I don’t understand the part of the sentence “the structure 1-6 is the with the highest similarity”, line 438
- In the Tables 2 to 6, the numbers of the compounds should be in bold, as it is in the in the text and Schemes.
- All the bold, italics, subscript, and superscript characters have to be included after the point 3.3. (end of page 6 and full page 7)
- What is the meaning of “m¬¬¬-“, line 319?
- There should be no “-“ in “corre-lation” line 311, “sub-stituent” line 322, “depend-ence” line 325, “struc-tures” line 332, “chap-ter” line 333, “pen-ta” line 339
Author Response
Dear Reviewer,
Thank you very much for the positive assessment and valuable comments provided. We have addressed all the comments as shown in the revised manuscript.
Sincerely, on behalf of all authors
Tomasz Siodła
In response to your comments:
- “In Schemes 1 and 2, there are structures in red, for which it is not obvious to consider them as para- (Scheme 1) or meta-derivatives (Scheme 2). An explanation is the text is needed.”
The following sentences have been edited/added in new line 87:
In para-type structures, interactions between donor and acceptor are inductive and resonance while in meta-type structures interactions are mostly inductive. Derivatives marked as red have different mutual orientation of substituents. It can be important in terms of e.g., field effect.
- “In the captions to Figures 4, 8, and 9, it should be indicated that the comparison is made with benzene derivatives.”
The mentioned captions have been clarified accordingly to the suggestion.
- “I don’t understand the part of the sentence “the structure 1-6 is the with the highest similarity”, line 438.”
The sentence has been corrected and clarified as below (new line 443):
(…) the structure 1-6 have the highest similarity to the m-trifluoromethyl-X-benzene in terms of accuracy and sensitivity.
- - I In the Tables 2 to 6, the numbers of the compounds should be in bold, as it is in the in the text and Schemes.
- All the bold, italics, subscript, and superscript characters have to be included after the point 3.3. (end of page 6 and full page 7)
- What is the meaning of “m¬¬¬-“, line 319?
- There should be no “-“ in “corre-lation” line 311, “sub-stituent” line 322, “depend-ence” line 325, “struc-tures” line 332, “chap-ter” line 333, “pen-ta” line 339.”
All editing errors have been tracked and corrected with particular emphasis on the bold, italics, subscript, and superscript characters.
Moreover, images and tables have been replaced alongside the text to fulfill the pages.
We have also added or edited the following sentences:
New line 162:
“From all the mono(CF3)substituted derivatives, structure 2-6 is the one with the highest value of R2 = 0.99. The substituent effect’s transmission in structure 2-6 is therefore very similar to that in p-trifluoromethyl-X-benzene (see Figure 2)”
New line 192:
“(…)it can be concluded that in terms of sensitivity, 1-4 substitution better resembles para-substituted benzene than 2-6”
New line 261:
“However, the sensitivity is lower due to the more distant substituents (a = 0.99 for 1-3 and 0.61 for 1-6).”
New line 263:
“The low value of R2 indicates disturbed interaction between substituents.”
We hope you will find our new manuscript to be sufficient for publication. Thank you!